# Numerical Modelling of a Dam-Regulated River Network for Balancing Water Supply and Ecological Flow Downstream

Yuxuan Gao [1], Wei Xiong [2,*] and Chenhao Wang [2,3]

1   Center of River and Lake Protection and Smart Water of Chengdu City, Chengdu 610000, China
2   School of Hydraulic and Environmental Engineering, Changsha University of Science & Technology, Changsha 410114, China
3   Changsha Shangyang Technology Co., Ltd., Changsha 410114, China
*   Correspondence: weixiong@csust.edu.cn

**Abstract:** Dam operation is regarded as an effective way to increase water, food, and energy security for society. However, with the increasing water demand and frequent extreme droughts, numerous rivers worldwide go through periods of water scarcity and water ecosystem deterioration to varying degrees. Balancing the water supply and ecological flow of the dam-regulated river network is essential in the context of river restoration. In this study, we proposed a hydrodynamic and water quality model of a dam-regulated river network balancing water supply and ecological flow using the Environmental Fluid Dynamics Code (EFDC). A section of Jinjiang watershed located in the southwestern of China was chosen as the study area. Firstly, the model was calibrated and validated. By comparing the simulated values with the measured values, the statistical analysis results showed that the relative root mean-squared error (RRMSE) values of water level, COD and $NH_3$-N were 5.5–8.1%, 23.6% and 28.4%, respectively, indicating an adequate degree of agreement between simulation and observation. Based on the established model, dam operation schemes under a dry hydrologic scenario and emergency contamination scenario were formulated to ensure the requirement of ecological water flow and water quality simultaneously. For the dry hydrologic scenario, the ecological water requirement could be satisfied through the dam operation. However, in an emergency contamination scenario, regional water quality requirements cannot be met through dam operation. The dam operation only plays a role in controlling the scope of pollution. This study is expected to provide scientific support for dam-regulated river network management and downstream river ecosystem protection.

**Keywords:** dam-regulated river network; ecological flow; hydrodynamic; numerical simulation; Environmental Fluid Dynamics Code (EFDC)





## 1. Introduction

Rivers are not only essential for socio-economic development, but also provide a habitat for various organisms and play an important role in maintaining ecological health [1]. Due to rapid economic development and population growth, water withdrawal for the economic water sector has increased dramatically, placing extreme pressure on ecological water use [2]. Nowadays, more than 60% of the world's rivers go through water scarcity and water ecosystem deterioration to varying degrees, and the situation will be further exacerbated by increasing water demand and frequent extreme droughts [3,4]. Therefore, determining how to use limited water resources to meet the targets of river ecological flow and other water uses is the top priority in the field of sustainable river management [5]. As defined in the Brisbane Declaration of 2018, environmental flows describe the quantity, timing, and quality of freshwater flows and levels necessary to sustain aquatic ecosystems which, in turn, support human cultures, economies, sustainable livelihoods, and well-being [6]. In China, the Ministry of Water Resources has issued documents on determining

the ecological flow of rivers and lakes and clarified the policies and measurements for determining and guaranteeing the ecological flow targets of rivers and lakes.

Dam and reservoir operations are the most common way to change the hydrological condition of rivers [7]. Changes in hydrological patterns affect a number of processes taking place in the river channel, including hydrodynamics, pollution transportation and diffusion, and biogeochemical processes [8]. According to the Global Reservoir and Dam Database, there are about 16.7 million of dam reservoirs larger than 0.01 ha in the world, with a total storage capacity of approximately 8070 km$^3$ [9]. Traditionally, the main purposes of dam operation are to increase water, food, and energy security for society [10]. However, this ignores the potential threats to the river ecosystem's health and stability caused by flow regime change [11]. The importance of the ecological flow in supporting stream functions during various seasons has been demonstrated in previous studies [12]. Over recent decades, extreme weather events, such as floods, droughts, and extreme temperatures, have occurred frequently around the world [13]. Therefore, much attention should be paid to ecological dam operation, which requires a comprehensive understanding of the impacts of dam operation on hydrodynamics and water quality, particularly in extreme weather conditions.

Numerous studies documented the effect of dams and reservoirs on streamflow and water quality. A numerical model is an efficient tool for comparing efficiency and visualizing spatial information, and can be applied in data-scarce areas [14]. Kim and Shin [15] constructed a numerical water quality model to investigate the influence of estuary dam operating conditions on water quality and algal bloom in the Yeongsan River. To ensure the water quality of the water source of the South-to-North Water Transfer Project, Chen et al. [16] constructed a three-dimensional eutrophication model for the Danjiangkou Reservoir and verified the model using DO, NH$_4$–N, TN, TP and Chl-a. Hua and Zhang [17] established a three-dimensional hydrodynamic and water quality model for the cascade reservoirs, which are the main drinking water sources in Shenzhen, China, to evaluate the performance of the water quality improvement measures. Yoshioka [18] proposed a simple but non-trivial numerical model of a dam–reservoir system to balance hydropower production and downstream environmental management. However, the current research on dam-regulated rivers has mainly concentrated on the hydrodynamic process and material transport of large dams or individual reservoirs; there has been little research on small cascade dams on urban river network.

Dam operation is regarded as an effective way to increase water, food, and energy security for society [19]. However, with the increasing water demand and frequent extreme droughts, numerous rivers worldwide go through water scarcity and water ecosystem deterioration to varying degrees. Determining how to balance the water supply and ecological flow of the dam-regulated river network is essential in the context of river restoration. Therefore, the objectives of this paper are (1) to establish a hydrodynamic and water quality model for the dam-regulated river network; (2) to analyse the relationships between cascade dam operation and river water level and quality; and (3) to formulate a dam operation scheme in a dry hydrologic scenario and an emergency contamination scenario to meet the requirements of water supply and ecological water flow simultaneously. This study aims provide scientific support for dam-regulated river network management and downstream river ecosystem protection.

## 2. Materials and Methods

### 2.1. Study Area

The study area is a section of the Jinjiang watershed located in southwestern China, which lies between 30°04′ N and 31°26′ N latitude and 102°54′ E and 104°53′ E longitude. Figure 1 shows the geographical location of the study area, it includes five major rivers, including the Fuhe River, Nanhe River, Huanhua River, Modi River, and Xijiao River, with a total length of about 29.5 km. The Fuhe River and Nanhe River are secondary tributaries of the Yangtze River; they are important channels in the central urban area of Chengdu,

Sichuan Province. There are seven rubber dams constructed in the Fuhe River and the Nanhe River. The details of the dams are shown in Table 1.

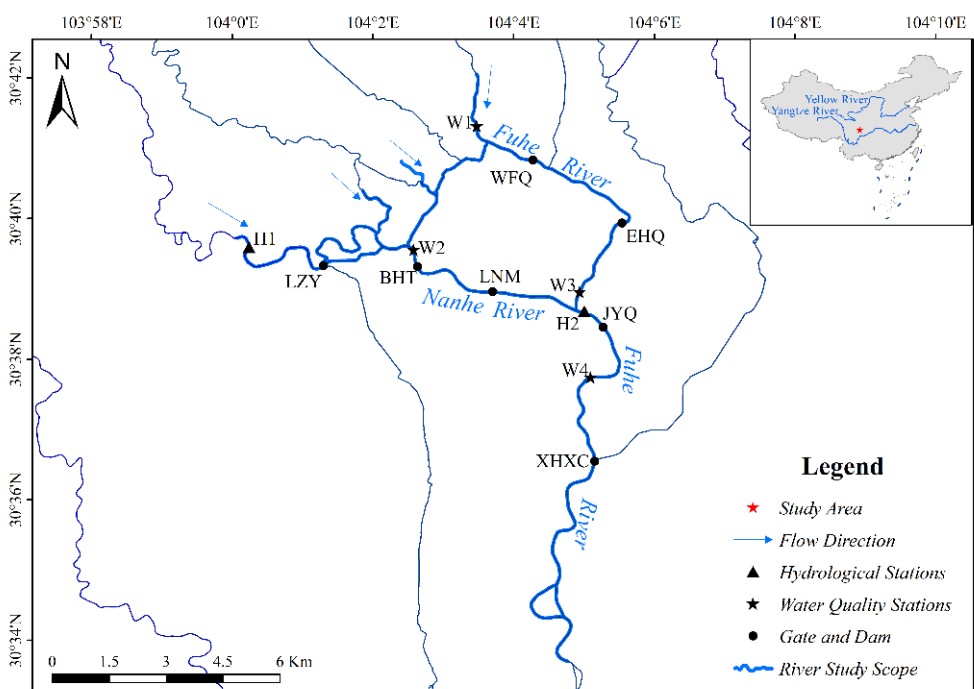

**Figure 1.** Schematic diagram for the study area.

**Table 1.** Basic information for the dams built in the study area.

| Name | Abbreviation | Longitude | Latitude | River |
|---|---|---|---|---|
| Wanfuqiao rubber dam | WFQ | 104°4′10″ E | 30°40′51″ N | Fuhe River |
| Erhaoqiao rubber dam | EHQ | 104°5′24″ E | 30°39′49″ N | Fuhe River |
| Baihuatan rubber dam | BHT | 104°2′22″ E | 30°39′35″ N | Nanhe River |
| Laonanmen rubber dam | LNM | 104°3′29″ E | 30°38′58″ N | Nanhe River |
| Longzhuayan rubber dam | LZY | 104°1′20″ E | 30°39′21″ N | Nanhe River |
| Jiuyanqiao rubber dam | JYQ | 104°5′12″ E | 30°38′33″ N | Fuhe River |
| Xiahexincun rubber dam | XHXC | 104°5′8″ E | 30°36′30″ N | Fuhe River |

Rubber dams adjust the height of the dam by filling and releasing air to control the upstream water level, thus playing a role in flood control, municipal water supply and tourism. Compared with conventional dams, rubber dams have characteristics such as low cost, short construction period, convenient management, and resistance to earthquake and wave impact.

The Jinjiang watershed is in a subtropical monsoon climate zone with a warm climate and abundant precipitation. The mean annual temperature is around 15.2–17.1 °C. The highest temperature occurs from July to August, with an average monthly temperature of 25.0–25.4 °C, while the lowest temperature occurs in January with an average monthly temperature of 2.4–5.6 °C. The mean annual precipitation in the Jinjiang watershed is 828.5–1265 mm, with the geographical distribution pattern decreasing from northwest to southeast. Temporally, precipitation in this region is mainly distributed from June to September, which accounts for about 60% of the annual precipitation. The runoff in the Jinjiang watershed is mainly recharged by precipitation, and the intra-annual variation in the runoff is consistent with the variation in the rainfall in the watershed. Water availability during the wet season (June to October) and the dry season (December to the following April) accounts for 63.9% and 20.7% of the total annual water resources, respectively. The lowest flow occurs in February and March, which accounts for only 5.99% of the total

annual water resources. According to the monitoring data from the hydrological station (H2 in Figure 1) from the years 1988–2017, the maximum monthly average flow was 77 m$^3$/s and occurred in July, while the minimum monthly average flow of 8.3 m$^3$/s occurred in March. According to the latest Chengdu Water Resources Bulletin, the water quality of the five rivers is relatively stable, and satisfied with the surface water quality standard for 'Grade III' as defined in the Environmental quality standards for surface water (GB 3838-2002) [20].

### 2.2. Data Description

Observed topological and bathymetric data, water depth and water level elevation data, water quality data, water withdrawal and effluent data were prepared for model generation and calibration. The topography of the rivers was designed on the basis of the field investigation, which was conducted by Chengdu Water Authority. The flow and water level of rivers were provided by gauging stations. The locations of water withdrawal and wastewater discharge were provided by the Chengdu Water Authority, while the quantity of water extraction and wastewater discharge were calculated based on population and per capita water use. Water quality data were collected from monthly monitor data from municipal or national water quality sites and from field measurement data. The measurements of water quality parameter concentrations in the laboratory followed the standard national methods [21]. The water quality variables included the chemical oxygen demand (COD) and ammonia nitrogen (NH$_3$-N), which are the two most important parameters of river ecosystems.

In this study, dam operation schemes were optimized under extreme conditions. Moreover, according to the hydrological data, the flow and water level in the study area is generally higher than the ecological water requirement during the wet season, and flood control is the main objective of the dam operation in this time period. Therefore, two scenarios called the dry hydrologic scenario and emergency contamination scenario were set. The flow and water level of rivers in the two scenarios were determined based on the 10th percentile of low daily average water level/flow. Based on the dry hydrologic scenario, the emergency contamination scenario assumed that an emergency pollution accident occurred at the entrance of the Nanhe River, with a pollutant discharge flow of 2 m$^3$/s. The COD and NH$_3$-N concentrations of the emergency discharge were assigned to 1000 mg/L and 10 mg/L, respectively. The ecological flow target used in this study was determined by the Sichuan Provincial Water Resources Department as 10% of the multi-year average flow at a given stream reach [22]. The ecological flow target is 30 m$^3$/s in the downstream of the Fuhe River. Meanwhile, the requirement of landscape/ecological water depth is 3 m downstream of the Nanhe River. In order to meet the water quality requirement of the water function zone, the river water quality should satisfy the surface water quality standard for 'Grade III'.

### 2.3. Model Description

Figure 2 summarises the methodological framework of the study. Step one includes the data collection and processing. The hydraulic structure data consist of time series of dam operation. In step two, the hydrodynamic and water quality model was configured and calibrated. In addition, based on the collected data, the regional ecological flow was determined and the satisfaction of ecological flow under the current dam operation was analysed. Step three was extreme condition scenarios setting. The simulation scenarios were set based on the analysis of ecological flow satisfaction, which focuses on the time period in which the ecological flow target cannot be satisfied. Based on the established hydrodynamic and water quality models, dam operation schemes were simulated under different scenarios. By evaluating and comparing the satisfaction of ecological flow, ecological water depth and water quality under different operation schemes, the optimal dam operation scheme was determined under different scenarios.

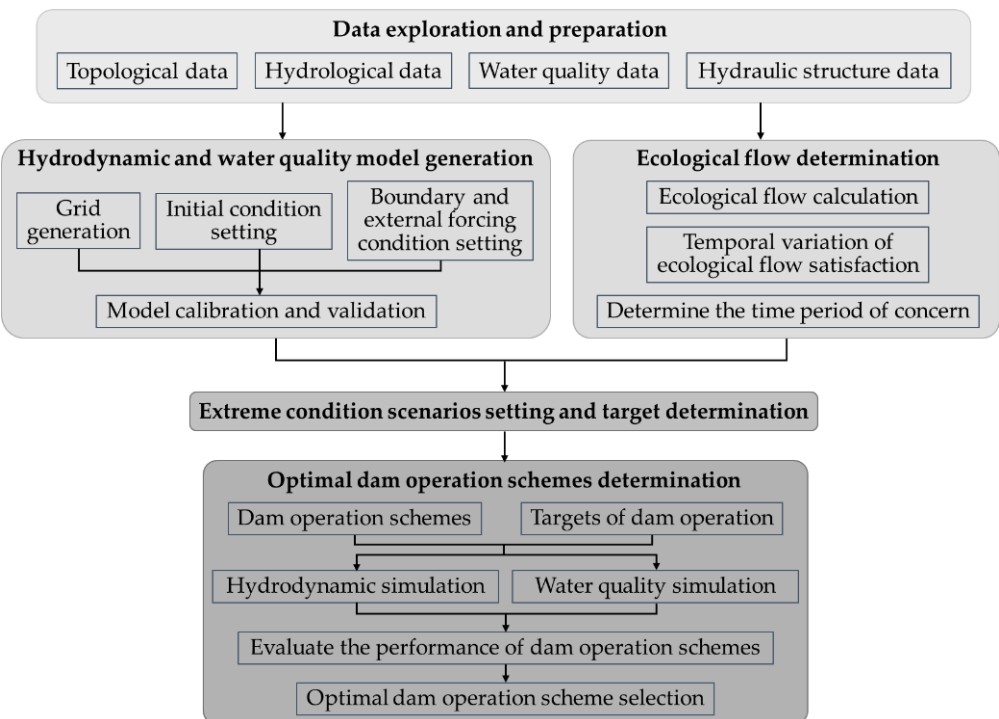

**Figure 2.** Methodological framework followed in the study.

In this study, a hydrodynamic and water quality model for the dam-regulated river network was developed using the Environmental Fluid Dynamic Code (EFDC). EFDC, initially developed by Hamrick in 1988, is a comprehensive modelling system recommended by the United States Environmental Protection Agency for surface water simulation [23]. It is one of the most widely used hydrodynamic and ecological models, which has been applied to simulate flow fields of water systems, material transport (temperature, salinity, and sediment transport), and ecological processes in rivers, reservoirs, lakes, wetland, and various aquatic systems [24–26].

The EFDC model composed of four key modules, which includes hydrodynamics, sediment transportation, water quality, and a toxic contaminant module [27]. The hydrodynamic model is the foundational module in EFDC, enabling the simulation of flow field, salinity, and temperature transport, while also providing essential hydrodynamic boundaries for the calculation of other modules [17]. The governing equation of the hydrodynamic model is a three-dimensional hydrostatic equation set, which comprises the continuity equation, Reynolds momentum equations and material transport equation [27]. In the formulated equations, second-order accuracy finite difference is adopted in the solution [28]. Since the curvilinear orthogonal grid performs well in adapting to complex boundary shape, and offers superior computational efficiency [29], a boundary-fitted curvilinear orthogonal horizontal coordinates and a sigma vertical coordinate are employed in the model. On the basis of the hydrodynamic results derived from the hydrodynamic module, the water quality module enables simulation of 8 categories of 21 water quality variables, including algae, phosphorus, nitrogen, chemical oxygen demand (COD), and so on [17]. Further details of the EFDC model are available in the literature [27,30].

## 3. Model Configuration and Calibration

The water surface of the five rivers in the study area was divided into 1142 columns and 973 rows of grid matrix, as shown in Figure 3a. The river surface covered 15,889 cells that have resolutions of 1.3 to 31.6 m for the i direction and 1.2 to 20.1 m for the j direction. The variation in grid size can precisely fit the boundaries of the river channels while ensuring computational efficiency. Considering the variations in the water level in rivers,

the model employed a dry–wet grid function to exclude calculations of water quality and quantity when the riverbed exposed. In this study, 0.1–0.15 m was chosen as the critical water depth of the dry and wet grid. The bottom elevation of the rivers in the study area was based on the observed river topography data (Figure 3b).

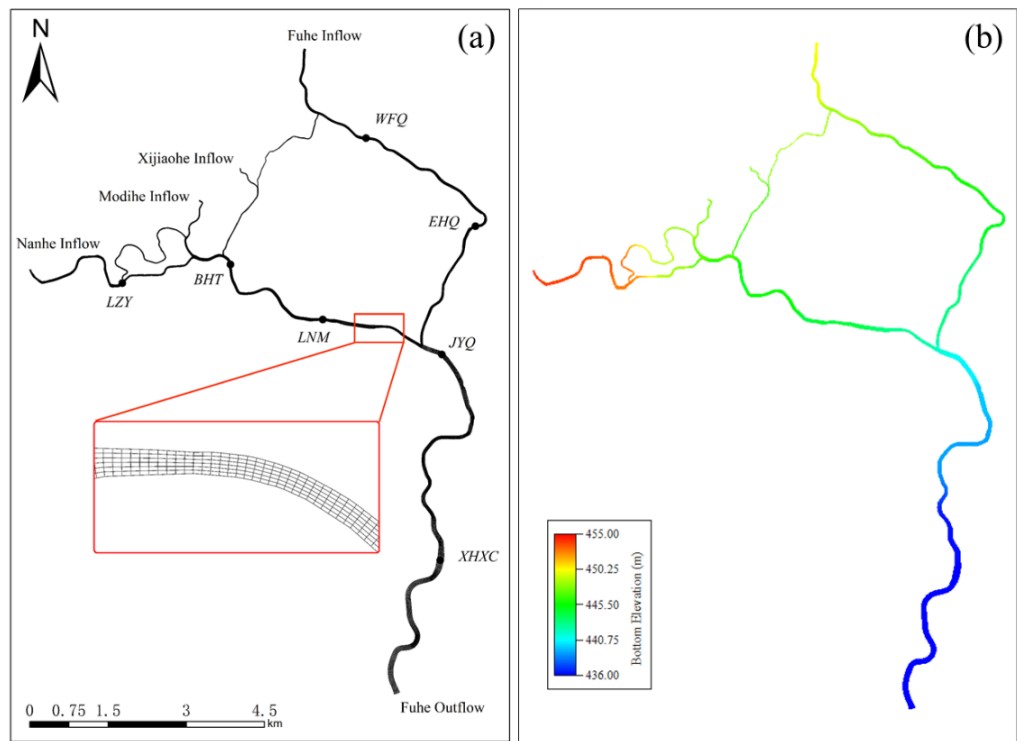

**Figure 3.** (**a**) Grid division and (**b**) bottom elevation of rivers in the study area.

Constant values were assigned to the initial conditions of water level, water temperature, and water quality, since Gao et al. [31] demonstrated that the effects of boundary conditions can eliminate the interference of initial conditions on simulation within a short period. The initial water level and water temperature of the rivers were determined based on the monthly average data from January to March at hydrological station H2 (Figure 1). Specifically, the initial water level and water temperature were assigned to 453.5 m and 8 °C, respectively. According to the latest Chengdu Water Resources Bulletin, the water quality of the five rivers in the study area exhibits a relatively stable condition, and satisfied with the surface water quality standard for 'Grade III' as defined in the Environmental quality standards for surface water (GB 3838-2002) [20]. Therefore, the initial concentration of COD and $NH_3$-N were assigned corresponding to the surface water quality standard for 'Grade III'.

External boundary conditions, such as inflow–outflow boundaries, water withdrawal and discharge boundaries, and hydraulic structures boundaries serve as driving forces for model simulations. For dry hydrologic and emergency contamination scenarios, the flow and water level of rivers were determined based on the 10th percentile of low daily average water level/flow. Nutrient loads (COD and $NH_3$-N) of inflow were obtained via field measurements from 1 January 2020 to 15 March 2020, which was recognized as an arid climate condition. Based on the dry hydrologic scenario, the emergency contamination scenario assumed that an emergency pollution accident occurred at the entrance of the Nanhe River, with a pollutant discharge flow of 2 $m^3$/s. The COD and $NH_3$-N concentration of the emergency discharge were assigned to 1000 mg/L and 10 mg/L, respectively.

The mean annual runoff of the Fuhe River was 39.85 $m^3$/s from the year 1988–2017, while the runoff of Fuhe River in 2020 was 35% less than the mean annual runoff. Therefore, we chose 2020 as the representative year for the model calibration. The model was

calibrated using the measured water level data from 1 January 2020 to 15 March 2020 at hydrological stations. The observed water level data at WFQ rubber dam during the same time period was used for the model validation. Model parameters were typically acquired through literature review [32–34], field measurements, and model calibration. The simulated values and observations of water level and water quality were compared to examine the model's performance. Correlation of determination ($R^2$), root-mean-squared-error (RMSE) and relative root mean-squared error (RRMSE) were used as the criteria for the optimal parameter selection.

## 4. Results and Discussion

### 4.1. Model Calibration

The sensitive hydrodynamic and water quality parameters were calibrated and the corresponding definitions and values are presented in Table 2. A dynamic time step was used to improve the calculation efficiency and the safety factor value was set to 0.1, ensuring that minimal numerical dispersion occurs during the calculation.

**Table 2.** Parameter-calibration results of the numerical model.

| Parameter | Definition | Value | Unit |
|:---:|:---:|:---:|:---:|
| ITERM | Maximum number of iterations | 1000 | - |
| $\Delta T$ | Time step | 0.3–10 | s |
| HDRY | Depth at which cell becomes dry | 0.1 | m |
| HWET | Depth at which cell becomes wet | 0.15 | m |
| ZROUGH | Bottom roughness | 0.035 | - |
| AHO | Constant horizontal diffusion | 1.0 | $m^2/s$ |
| AHD | Dimensionless horizontal momentum diffusivity | 0.2 | - |
| AVO | Background molecular eddy (kinematic) viscosity | 0.001 | $m^2/s$ |
| ABO | Background molecular diffusivity | $1 \times 10^{-8}$ | $m^2/s$ |
| KCOD | COD concentration decay rate | 0.0020 | 1/day |
| MAKN | Maximum nitrification rate | 0.007 | 1/day |
| $OHSCNH_4$ | $NH_4$ Half-Saturation Constant for Nitrification | 0.3 | $gN/m^3$ |
| $OHSCNO_3$ | $NO_3$ Half-Saturation Constant for Denitrification | 0.1 | $gN/m^3$ |
| RTN | Reference temperature for Nitrification | 27 | $^\circ C$ |

The simulated water levels at hydrological stations H1 and H2 were compared with the observations to evaluate the model performance, as shown in Figure 4. At Xiahexin-cun hydrological station (H1), the R2, RMSE, and RRMSE were 0.90, 0.034 m, and 8.1%, respectively. At Lonanmen hydrological station (H2), the R2, RMSE and RRMSE were 0.88, 0.053 m and 5.5%, respectively. The results showed that the simulated water values were well matched with the observations, indicating a good performance of the hydrodynamic model. Furthermore, monitoring water level data at WFQ rubber dam were used for the hydrodynamic model validation (Figure 1). The R2, RMSE and RRMSE were 0.89, 0.304 m and 16.0%, respectively. The validation process has enhanced the model's reliability, providing a satisfactory foundation for water quality simulation.

Based on the hydrodynamic model that has been calibrated and validated, calibration of the water quality model was performed. There are four water quality stations established within the study area (Figure 1), and each station conducts measurements and updates data every 4 h. Since monitoring data of the water quality station W1 and W2 were utilized as the external boundary in the model, they were excluded from the model calibration. Monitoring data obtained from the water quality station W3 and W4 were utilized for the model calibration. Key parameters of the water quality model were calibrated and listed in Table 2. For water quality simulation performances, the statistical analysis results show that the RRMSE values of COD and $NH_3$-N were 23.6% and 28.4%, indicating an adequate degree of agreement between simulation and observation.

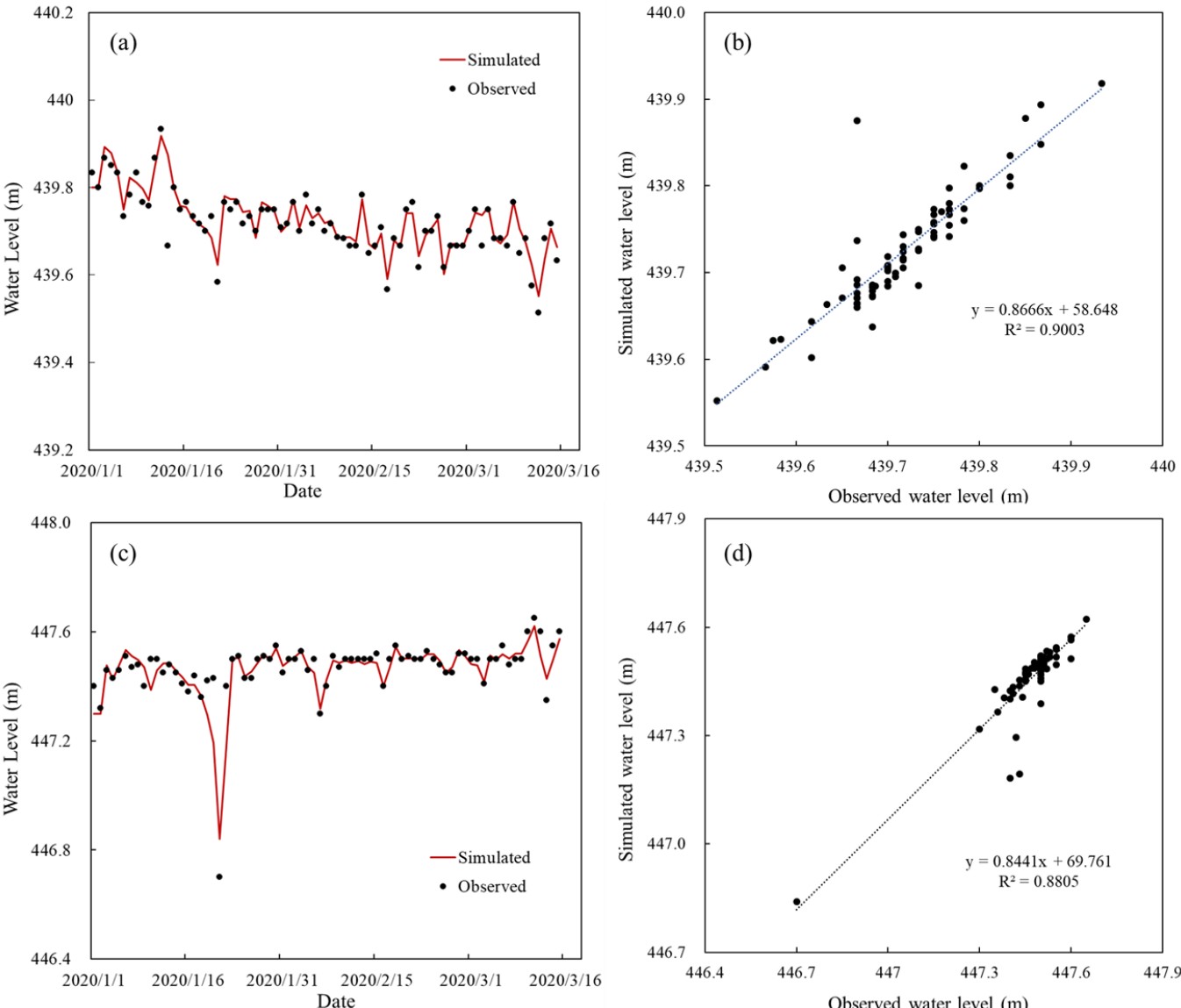

**Figure 4.** Observed and simulated values of water level at (**a**,**b**) Xiahexincun and (**c**,**d**) Laonanmen hydrological stations.

### 4.2. Dry Hydrologic Scenario

According to the requirements of the Sichuan Provincial Water Resources Department, the ecological flow target is 30 m$^3$/s downstream of the Fuhe River. Meanwhile, the landscape/ecological water depth requirement is 3 m in the downstream of the Nanhe River. To ensure the requirement of ecological water flow and depth simultaneously, two dam operation schemes were formulated under dry hydrologic scenario. For both operation schemes, the height of the BHT and LNM rubber dams were increased to raise the downstream water level of the Nanhe River, while the height of the WFQ and EHQ rubber dams decreased to ensure the ecological flow requirement of the Fuhe River. For operation scheme 1, the height of the BHT and LNM rubber dams was set as 2.8 m, and the height of the WFQ and EHQ rubber dams were set as 1 m. For operation scheme 2, the height of the BHT and LNM rubber dams was set as 3.5 m, and the height of the WFQ and EHQ rubber dams was set to a flat dam. The water depth variation at the control cross-section (104°4′58″ E, 30°38′35″ N) downstream of the Nanhe River under operation schemes is shown in Figure 5.

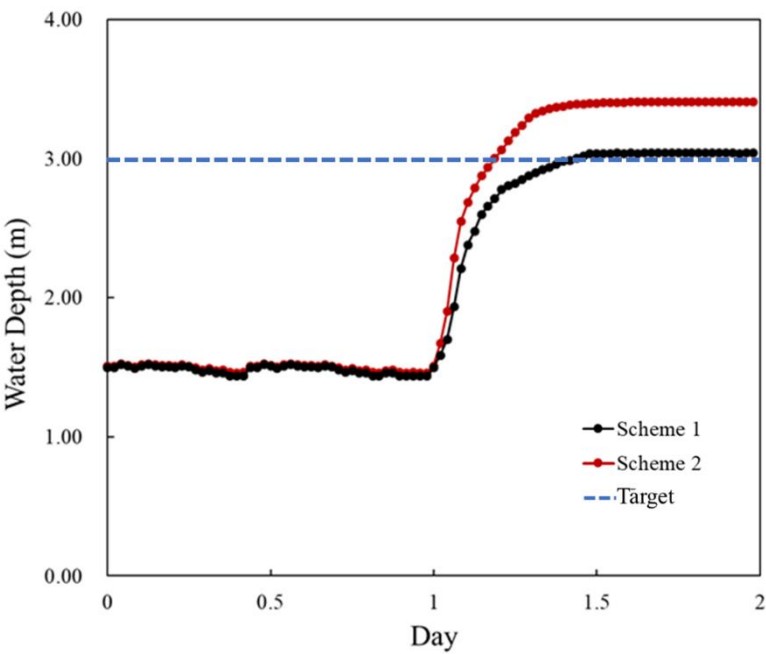

**Figure 5.** Water depth variation at the control cross-section (104°4′58″ E, 30°38′35″ N) downstream of the Nanhe River in different operation schemes under dry hydrological scenario.

The initial downstream water level of the Nanhe River was 1.51 m and the operation scheme 1 was implemented on day 1. After that, the water depth in the downstream of the Nanhe River gradually increased to 3.03 m, corresponding to a water level of 447.53 m. In the case of operation scheme 2, after the operation scheme was implemented, the water depth downstream of the Nanhe River was finally stabilized at 3.41 m, corresponding to a water level of 447.91 m. Compared with operation scheme 1, the water level had a faster response in scheme 2. For operation scheme 1, it took 10.5 h for the control cross-section to reach a water depth of 3 m, while in the case of scheme 2, it took only 4.5 h to meet the water depth requirements. It can be seen that under dry hydrological conditions, the ecological water requirement could be satisfied through the dam operation.

### 4.3. Emergency Contamination Scenario

On the basis of the dry hydrologic scenario, it was assumed that an emergency pollution accident occurred at the entrance of the Nanhe River, with a pollutant discharge flow of 2 m$^3$/s. The COD and NH$_3$-N concentration of the emergency discharge were assigned to 1000 mg/L and 10 mg/L, respectively. In order to dilute the contaminants and improve the self-purification capacity of the river, the heights of the LZY and BHT rubber dams on the Nanhe River decreased. The height of the LZY and BHT rubber dams was set as 0.5 m in operation scheme 1, and 1 m in operation scheme 2. Meanwhile, the height of the JYQ rubber dam was set as 3.5 m to ensure the downstream water level of the Nanhe River, as well as to control the scope of contamination. The water quality at the control cross-section (104°4′58″ E, 30°38′35″ N) downstream of the Nanhe River under the operation schemes is shown in Figure 6.

For both operation schemes, the maximum COD concentration occurred at the fourth hour, corresponding to 150 and 132 mg/L in schemes 1 and 2, respectively. For NH$_3$-N, the maximum concentration also occurred at the fourth hour, corresponding to 2.0 and 1.8 mg/L in schemes 1 and 2, respectively. The results indicated that it takes approximately 4 h for pollutants to transport from the emergency discharge to the control cross-section. Due to the water inflow from the upstream of the Nanhe river and Fuhe river, the pollutant concentration decreased rapidly in a short period of time. Compared to scheme 2, the pollutant concentrations at the downstream outlet cross-section were greater in scheme 1 between 4 and 20 h. This is because the flow rate of Nanhe River was larger in scheme 1

than in scheme 2, resulting in faster pollutant transport in scheme 1. The faster the pollutant transport, the shorter the residence time of the pollutants in the river. This may lead to a greater pollutant concentration in the downstream outlet cross-section in scheme 1 than in scheme 2 in a short period of time. After 20 h, the pollutant concentrations in scheme 1 were lower than in scheme 2. However, the COD concentration was about 80 mg/L after 48 h under both schemes, which was much higher than the surface water quality standard for 'Grade III'. The result indicated that when emergency discharges occur during dry periods, regional water quality requirements cannot be met through dam operation.

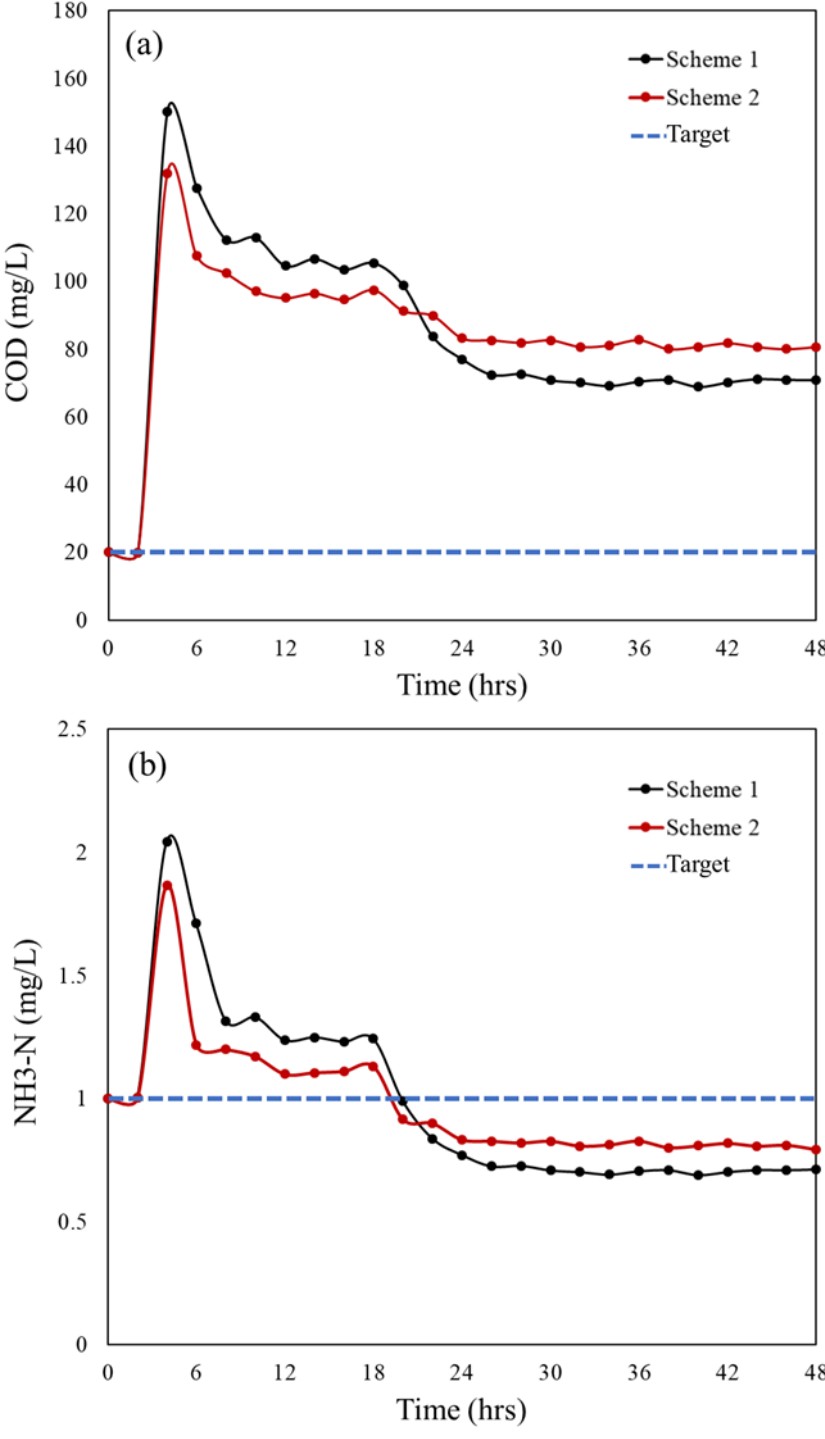

**Figure 6.** (**a**) COD and (**b**) NH$_3$-N concentration in the downstream of Nanhe River in different operation schemes under emergency contamination scenario.

*4.4. Limitations*

There were many constraints related to the dam-regulated river network model established in this study. First of all, the model calibration period and validation period were short, which affected the reliability of the model. Additionally, since the model was not combined with the runoff forecasting model, the model can only train according to previous data to determine the parameters. Thus, the model has poor extensibility. For the same reason, instead of optimizing the dam operation scheme for a possible future scenario, two extreme condition scenarios are simply assumed and studied in this paper. For the performance evaluation of the dam operation schemes, this paper only focused on the satisfaction of ecological flow, ecological water level, and water quality at limited control cross-sections. Therefore, the performance of the dam operation schemes has not been fully evaluated, which means the selected optimal dam operation scheme could result in a significant negative impact on ecological flows at other locations. In the circumstance in which none of the dam operation schemes can meet the water supply and ecological water flow targets simultaneously, inter-basin water transfers or reclaimed water for ecological recharge should be considered. Another obvious limitation is that the ecological flow target used in this study was determined as 10% of the multi-year average flow at a given stream reach. Using a simple ecological flow target may not be protective of the overall stream ecosystem health. Future studies could utilize more holistic environmental flow recommendations using a functional flows approach that determines ecological flows that are hypothesized to support stream functions [12,35].

**5. Conclusions**

This study established a hydrodynamic and water quality model of a dam-regulated river network balancing water supply and ecological flow based on the EFDC. A section of the Jinjiang watershed located in southwestern China was chosen as the study area. The simulated values and observations of water level and water quality were compared to examine the model performance, and the results showed an adequate degree of agreement between simulation and observation. Based on the established model, dam operation schemes in a dry hydrologic scenario and an emergency contamination scenario were formulated to ensure the requirement of ecological water flow and water quality simultaneously. In the dry hydrologic scenario, the ecological water requirement could be satisfied through the dam operation. However, in the emergency contamination scenario, regional water quality requirements could not be met through dam operation. The dam operation only plays a role in controlling the scope of pollution. The modelling framework developed in this study can be adapted in other dam-regulated rivers across the globe for the dam operation scheme formulation under extreme conditions to ensure the requirement of water supply and ecological water flow simultaneously. This study is expected to provide scientific support for dam-regulated river network management and downstream river ecosystem protection.

**Author Contributions:** Conceptualization, Y.G.; methodology, W.X.; data curation, Y.G.; investigation, C.W.; data curation, Y.G. and C.W.; writing—original draft preparation, W.X. and Y.G.; writing—review and editing, W.X. and Y.G. All authors have read and agreed to the published version of the manuscript.

**Funding:** A Project Supported by Scientific Research Fund of Hunan Provincial Education Department (22B0331) and the Major Water Conservancy Science and Technology Projects of Hunan Province (No. XSKJ2021000-07).

**Data Availability Statement:** Data is unavailable due to privacy or ethical restrictions.

**Acknowledgments:** The authors are grateful for the help from Cao Jinsong in the article's writing and revision, and for the help from Zhang Kai with the data processing. They are also grateful to the associate editors and reviewers for their helpful comments.

**Conflicts of Interest:** The authors declare no conflict of interest.

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
