# Peer review of "Numerical Modelling of a Dam-Regulated River Network for Balancing Water Supply and Ecological Flow Downstream"

_water, doi:10.3390/w15101962_

Round 1

Reviewer 1 Report

Overall comments:

Overall, this paper is well written and the work is logical.  However, this paper is lacking a discussion that touches on the broader implications of the study.  What is the contribution to the wider field? Are there general guidance or a framework that you can provide to inform future dam operations within this watershed and to other similar contexts?  Elaborate on what this paper is providing beyond specific recommendations for the two scenarios – which may not happen in real life.  Also, is there a baseline scenario that can be evaluated to show how current dam operations are doing compared to the explored operation scenarios?  It was unclear as to the point of the two scenarios that are being evaluated and the two operations.  How could someone that is unfamiliar from this study benefit from this study?  This paper would be greatly improved by looking at a wider range of future scenarios beyond just the 10th percentile of flow/depth.  Perhaps you can provide a modeling framework to determine optimal dam operation schemes under variable flow conditions and then test or apply that for the contaminant spill scenario.  Providing a modeling framework to determine optimal dam operations would provide useful to make management decisions that benefit both hydrology and water quality.   Additionally, I have a major concern about the short calibration period (3 months) and essentially no independent validation data.  The authors should reserve a subset of the observed data from the calibration to do an independent model performance evaluation for this to be a publishable study.  The authors should add a section on the model limitations and provide suggestions for improvement. 

Specific Comments:

-        Ln 40: spell out environmental flows (e-flows)

-        Ln 47: patterns instead of rhythms

-        Ln 56: the wet and dry-season matters.  I wouldn’t reduce the ecological flow needs to just the dry-season. See paper on functional flows by Yarnell et al 20XX and 2021.  Please acknowledge the importance of multiple aspects of the annual hydrograph in terms of stream functions that various seasonal flows support and justify why you are solely focusing on the dry-season (which appears to be driven by water quality concerns).

-        Figure 1: it is hard to see the labels and some of the overlapping stars on top of the point dams

-        Ln 131: please justify why and how these two scenarios were selected. Why not simulate a wet, moderate, and dry scenario?  

-        Ln 135: please briefly explain how these flow targets were developed.  Were they based on the flow needs of a specific species life stage?  Is there a study or report that can be cited?

-        Ln 137: I would reword “water function of rivers”.  You could say, “In order to meet the surface water quality standard…”.  There isn’t much evidence that meeting these targets would support functions of the river.  Also, could you describe why the Grade III standard was selected and what that means?  Either here or on line 115.

-        Ln 142: specify what the EPA recommended this model for – dam applications?

-        Ln 189: you should describe the emergency contaminant scenario when you introduce the two scenarios above (Ln 131).

-        Ln 196: The calibration period is very concerning because 1) there are only three months of observed data for calibration, 2) are these three months considered the dry season?, and 3) it seems as if you did not have an independent validation dataset to truly assess model performance outside of your calibration dataset.  Please explicitly address my concerns in the text and clarify if 2020 is a representative year.  Was this a drought year and you chose this time period because you are looking at the extreme?  Provide justification.  I would also recommend calling out the dry season months in the study area description (~ln 113).

-        Ln 211: again, it seems as if the maximum error occurred during the same calibration period (January 2020). Please include a model limitation section where you call out that there was not an independent validation datasat, if this is the case.

-        Figure 3: It would be more informative to see the observed vs predicted water levels instead of the timeseries of water level.  It’s hard to tell if there is bias in the model predictions, as is.

-        Figure 4: could you indicate the water level target in this plot?

-        Figure 5: could you also indicate the water quality standard as a line on this plot so you can see when and if the concentrations satisfy the water quality target?

Author Response

Dear Reviewer #1,

We are grateful for your time and effort in correcting the manuscript. We have carefully checked our manuscript. Your questions and comments have been taken into full account in the new revised manuscript and was marked up using the “Track Changes” function.  Please find the details in the upload file.

Reviewer 2 Report

The paper investigates the water supply related to ecological flow of a dam-regulated river network applying the Environmental Fluid Dynamics Code, a hydrodynamic and water quality model for the dam-regulated river.

The manuscript can be judged to add knowledge in its topic and it adheres to the “Water” standards.

On the other hand the text current form needs some clarification:

- The hydrography illustrated in figures 1 and 2 is not clear: it seems that the rivers touch each other, forming a sort of closed quadrilateral channel! Which is obviously impossible for physical reasons.

- Lines 176-181: the choice of boundary conditions has to be better explained and justified.

- Text referring to “water level” and figure 3-a: it is necessary to explain how the authors estimated by the model this parameter, because it requires a specific knowledge of the riverbed morphology.

- Lines 236-249: dam management scenarios need to be better clarified and discussed. It seems that the authors assume that the water levels in the dams can be freely varied, to ensure the ecological flow target of 30 m3/s. But the question is: in dry conditions is there always enough water in the dams to release the target flow? Without this analysis, the study remains theoretical, although it has an undoubted interest. Consequently, this question should have space in the discussion of the paper.

Author Response

Dear Reviewer #2,

We are grateful for your time and effort in correcting the manuscript. We have carefully checked our manuscript. Your questions and comments have been taken into full account in the new revised manuscript and was marked up using the “Track Changes” function.  Please find the details in the upload file.

Round 2

Reviewer 1 Report

The authors did an excellent job addressing my comments and improving the overall quality of the paper.  With these revisions, and addressing the minor comments below, I accept the publication of this manuscript.

Ln 168 to 173:  These sentences are misleading, as you are suggesting that the ecological flow used in this study was based on a functional flows approach.  This is untrue, as it states that the ecological flow target was 10% of the multi-year average flow of the cross-section.  Please remove the entire first sentence of this paragraph.  I would reword these lines to something like, “The ecological flow target used in this study was determined by the Sichuan Provincial Water Resources Department as 10% of the multi-year average flow at a given stream reach [reference]”.

Ln 398: Although the point of the study was not to develop ecological flow targets, but use them to inform dam operations, you should point out that using a simple ecological flow requirement, such as 10% of the observed mean annual flow, may not be protective of the overall stream ecosystem health.  If you look into the literature, 10% of unimpaired flows could be protective, but there is no scientific evidence that 10% of actual/observed flows are protective (for example, if actual flows are lower than natural due to diversions, 10% of flows could mean rivers run dry).  Future studies could utilize more holistic environmental flow recommendations using a functional flows approach that determines ecological flows that are hypothesized to support stream functions (Yarnell et al; and Stein et al).

Ln 430: You should also mention that the modeling framework developed in this study can be adapted and used in other dam-regulated rivers across the globe.

Author Response

Dear Reviewer #1,

We are grateful for your time and effort in correcting the manuscript. We have carefully checked our manuscript. Your questions and comments have been taken into full account in the new revised manuscript and was marked up using the “Track Changes” function.

Reviewer 2 Report

Accept in present form

Author Response

Dear Reviewer #2,

We are grateful for your time and effort in correcting the manuscript. Your comments are valuable to improve our manuscript. Thanks a lot.